# Bio-Ethology of *Vespa crabro* in Sardinia (Italy), an Area of New Introduction

**DOI:** 10.3390/biology11040518

**Published:** 2022-03-28

**Authors:** Michelina Pusceddu, Matteo Lezzeri, Arturo Cocco, Ignazio Floris, Alberto Satta

**Affiliations:** Department of Agricultural Sciences, University of Sassari, Viale Italia 39, 07100 Sassari, Italy; mpusceddu@uniss.it (M.P.); mlezzeri@uniss.it (M.L.); acocco@uniss.it (A.C.); albsatta@uniss.it (A.S.)

**Keywords:** European hornet, intranidal and extranidal tasks, foraging activity, prey preference, predation pressure

## Abstract

**Simple Summary:**

Alien insects, including hornets, may show invasive traits in non-native areas, thus threatening the ecological balance of natural and agro-ecosystems. The European hornet, *Vespa crabro*, is an omnivorous eusocial insect predator of many arthropods, including honey bees. It is native to Eurasia and established in Sardinia (Italy) in 2010, where it is an alien species. *Vespa crabro* does not represent a danger for beekeeping in its native area, although the potential environmental impacts in non-native areas are unknown. In view of the lack of such studies, this research investigated the potential invasive traits of *V. crabro* in an area of new introduction, with special regard to predatory activity against honey bees. Observations of hornet foraging behaviour in open fields highlighted a prevalent foraging activity on fruits and no preference for bees compared to other recognizable arthropods. Furthermore, behavioural and predatory observations of *V. crabro* near apiaries showed that foragers returned to nests carrying preys in 1% of cases. *Vespa crabro* did not show invasive traits nor notable behavioural changes in a non-native compared to its native area, as the hornet was confirmed to be a mild predator of honey bees. Therefore, the European hornet is not likely to represent a threat to beekeeping in Sardinia.

**Abstract:**

*Vespa crabro*, also known as European hornet, is a eusocial Vespidae originally from Eurasia that was accidentally introduced on the island of Sardinia (Italy) in 2010. Currently, its distribution is limited to the northern area of the island. Considering that a non-harmful species in its native region can exhibit invasive behaviour when established in new environments, bio-ethological observations were conducted to better understand whether *V. crabro* could show invasive traits in Sardinia, which represents a new introduction area. Data on the foraging activity of the European hornet in open fields were collected within a citizen science monitoring program carried out in Sardinia from 2018 to 2020. Moreover, specific behavioural observations were made in apiaries to assess the predatory activity of the hornet towards honey bees and at the entrance of free-living European hornet colonies to evaluate other aspects of its behaviour, i.e., intranidal and extranidal tasks. The results of our study are discussed in relation to the behavioural traits known for this species in its native areas to place the behavioural repertoire of *V. crabro* in Sardinia into a wider context. Our observations revealed that *V. crabro* did not show any changes in behavioural traits in Sardinia compared to those described in its area of origin, so the risk of becoming an invasive species on this island seems unlikely.

## 1. Introduction 

The European hornet, *Vespa crabro* L. (Hymenoptera: Vespidae), is a social insect native to Eurasia [1], also present as an alien species in Central and North America [2,3], including Canada [4,5]. In 2010, *V. crabro* was accidentally introduced to Sardinia (Italy), where its distribution is currently limited to the northern area of the island [6]. 

As observed in other hornet species, such as *Vespa orientalis* L. and *Vespa velutina* Lepeletier (Hymenoptera: Vespidae), the colony life cycle of *V. crabro* can be summarized in three steps: (1) nest foundation in spring, when queens, previously fertilized in autumn, leave their overwintering sites; (2) colony expansion from summer until autumn, when new potential founding queens and males are produced; (3) colony decay from late fall onwards [7,8,9,10]. Usually, *V. crabro* constructs large paper nests in closed sites, mainly in hollow trees, but also in abandoned beehives, hollow walls, and other building cavities [7,8,11]. Like other wasps, *V. crabro* is a predator of a wide range of arthropods [7,11,12], including odonates [13], cicadas [14], and honey bees [15]. In Japan and North America, *V. crabro* is considered a semi-generalist species as it hunts a variety of prey species with a preference for cicadas. Under laboratory conditions, this species showed a preference for honey bees over other insect prey and non-prey protein sources [16]. However, *V. crabro* does not represent a danger for beekeeping in its native area, although it can have a considerable impact on weakened colonies [15]. This is because in the evolutionary arms race both European hornet and Western honey bee have evolved various attack and defence strategies, respectively. The hornet’s main hunting strategy is attacking forager bees in flight to proximity of hive entrances when they return to the hive tired and burdened by the load [15]. Rarely, hornets attack bees at the hive entrance, where bees are most likely to respond with social defence, such as heat balling [15]. The food requirements of a *V. crabro* colony are influenced by the phase of colony expansion. The nutritional demand for protein increases from summer to fall, as the brood develops and future founding queens are produced [10,11,12]. 

The response of a species to a new environment varies greatly, and a non-harmful species in its native region can exhibit invasive behaviour in areas of new introduction [17]. The reasons lie in the favourable climate conditions of the new environment, availability of nesting sites, e.g., ancient olive groves present in Sardinia, and the lack of native competitors, such as other hornet species, that could facilitate the dispersal of the invader [17]. For example, the German wasp *Vespula germanica* (F.) (Hymenoptera: Vespidae), a mild predator of *Apis mellifera* L. (Hymenoptera: Apidae) in its native range [18,19], has become an important threat to beekeeping and biodiversity in areas of new introduction of the wasp [20,21]. Given the lack of information on *V. crabro* as an alien species and the need to protect the balance of sensitive island ecosystems [6], we conducted our study to better understand the bio-ethology of this species in a new introduction area. To achieve this goal, we analysed data on the feeding behaviour of *V. crabro* provided by citizen scientists involved in a monitoring program for the European hornet conducted in Sardinia from 2018 to 2020. In addition, specific behavioural observations were made in apiaries to assess the predatory activity of the European hornet towards honey bees and in free hornet colonies to determine intranidal and extranidal tasks of the queen and workers and the daily foraging activity of the colony. Finally, to place the behavioural repertoire of *V. crabro* detected in an area of new introduction (Sardinia) into a wider context, we discussed the results of our study in relation to the behavioural traits known for this species in its native areas.

## 2. Methods

### 2.1. European Hornet Foraging Activity

As part of the Interreg ALIEM project “*Action pour Limiter les risques de diffusion des espèces Introduites Envahissantes en Méditerranée*”, a monitoring program of the European hornet was conducted in Sardinia (Italy) from 4 June 2018 to 22 June 2020 using verified citizen science [6]. A total of 179 citizens participated in the study. For each nest or individual of *V. crabro*, the reports compiled by volunteer citizens included: date and place (geographic coordinates) of observation, habitat type (rural or urban), and environmental sub-category (forest, agricultural, apiary, building, garden, or other). Recruited volunteers were also required to provide, through photos or videos, additional information concerning the activity performed by the observed hornet, i.e., predation of honey bees or other prey, foraging on flowers, fruits, water, or sugary exudates. This additional information was taken into account in this paper. Further details on recruitment and evaluation of volunteers and data collection are reported elsewhere [6].

### 2.2. Predation Pressure of the European Hornet on Hives

Specific observations in apiaries were made at three sites, located in Cugnana, Olbia Santa Lucia (Olbia S.L.) and Telti (sites no. 1, 2, and 6, respectively) in northern Sardinia (Italy) (Table 1), to evaluate the predation pressure exerted by hornet queens or workers on hives, following the same method described by Pusceddu et al. [18,19]. Briefly, attacks by hornets were visually monitored for one day, between 9:00 A.M. and 5:00 P.M., when wasps are known to visit apiaries [15]. Two operators, working simultaneously, recorded the number of attacks along a transect traced in front of the hives. Each operator surveyed half of the hives present in the apiaries. The apiaries in sites no. 1 and 6 (Cugnana and Telti) consisted of 40 and 24 hives, respectively, and the nearby hornet nests were constituted only of the queen and the brood, located in the proximity of some hives (Table 1). The apiary in site no. 2 (Olbia S.L.) was composed of 15 bee colonies placed 500 m far from a hornet nest that included the queen and workers (Table 1). Therefore, the predatory pressure of the queen was assessed in the first two apiaries, whereas the predatory activity of the workers was assessed in the third apiary.

### 2.3. Observations on European Hornet Free-Living Colonies

During 2018, 2019 and 2020, observations were made on six *V. crabro* nests which had been identified following the reports provided by citizen scientists in Sardinia (Figure 1). A description of the characteristics of the observed nests and the type of data recorded is given in Table 1.

Two of these nests consisted of the queen and brood (capped and uncapped) only (nests no. 1 and 6 in Cugnana and Telti, respectively), whereas the other four nests included also workers (nests no. 2–5 in Olbia S.L., San Teodoro, Siniscola, and Olbia Santa Mariedda (Olbia S.M.), respectively). Four nests, including both nests consisting of the queen and brood only, were within or in proximity to apiaries. All observations were made during the period in which the foraging and nest activities of *V. crabro* were more intense (June, July, August, and September). Videos were recorded with an HD video camera (Canon LEGRIA HF R506, Tokyo, Japan) placed approximately 30 cm away from the nest entrance, within the time range between 6:00 A.M. and 8:00 P.M. (Appendix A).

Subsequently, two operators, working independently of one another, screened the recorded videos using a slow-motion system (VLC software v.3.0.12) and noted the frequencies of the behaviours observed, i.e., the number of events per unit of time, as described in the ethogram (see the Section 3.3.1).

In nests no. 1 and 6, consisting only of the queen and brood, we used the focal animal sampling method [22] to survey the tasks performed by the queen inside the nest on the first day of observation, when the entry hole was large enough to allow us to observe what was occurring inside the nest. To assess nest development, the number of uncapped and capped brood cells was counted at the beginning of the observations. Intranidal and extranidal queen activity rates were noted by recording queen entries and exits and the time spent by the queen in and out of the nest, as well as the interactions between the queen and hornet larvae and any other species that visited the nest. In nest no. 6, the hornet nest entrance context was observed for one day, whereas observations continued for five days at nest no. 1.

Bio-ethological observations were also made on four free-living *V. crabro* colonies consisting of the queen, uncapped and capped brood, and workers (nests no. 2–5) (Table 1) by using the recording technique described above. The daily foraging activity of the free-living colonies was estimated during a three-day observation period and the number of incoming and outgoing foragers was reported following the method described by Perrard et al. [23]. The flow of foragers, an estimate of the colony size [8], allowed evaluation of the development of the colony without having to destroy the nest. The type of load carried between the jaws by the incoming foragers was classified into the following categories: plant material (building material), prey, pellets (unidentifiable food processed elsewhere), and nothing between the mandibles (this category does not exclude the collection of water or sugary exudates). The relative frequencies of the type of load carried by the incoming foragers and the type of prey were calculated as the ratio between the frequency of a particular event and the total frequency of all events multiplied by 100. Since the first comb of nest no. 4 was visible from the outside, we detected the tasks performed by workers inside that nest by using the all-occurrences sampling method [22]. The interactions between workers and larvae and other species that visited the nest were also noted. These behaviours complemented the queen ethogram. In addition, the hornet ethogram was supplemented with further information reported in the literature for similar species in order to underline the bio-ethological evolutionary aspect.

### 2.4. Data Analysis

A chi squared test was used to measure the proportional difference in intranidal and extranidal activities of the European hornet queen between nest no. 6 (Telti) and no. 1 (Cugnana), which had similar nest size and habitat (rural). To reduce the chances of a type I error, continuity correction was also used for the chi-squared test when the sample size was less than 200 [18,24]. The Wilcoxon rank sum test (unpaired comparisons) was used to compare the average duration of each trip and the time spent inside the nest between two consecutive flights between Telti and Cugnana nests. Differences in the proportion of payloads carried by foragers were investigated by analysis of variance followed by Tukey’s post hoc test. Percentage data were arcsine-square root transformed prior to analysis to meet the assumption of homogeneity of variance, which was tested by Bartlett’s test (*p* > 0.05). All tests were carried out using R v3.02.

## 3. Results

### 3.1. European Hornet Foraging Activity

During the three-year monitoring program of European hornets in north-eastern Sardinia, we received a total of 265 valid reports (101 in 2018, 68 in 2019, and 96 in 2020) from volunteers. Considering only those in which *V. crabro* was engaged in foraging activities (39 photos and/or videos), we found that hornets were reported feeding on ripe fruit (*Ficus carica* L., *Pyrus communis* L., *Vitis vinifera* L., *Citrullus lanatus* (Thunb.) Matsum. and Nakai) in 20 cases (51.3%), feeding on sugar exudates in seven cases (17.9%), feeding on flowers (Eucalyptus camaldulensis Dehnh., Opuntia ficus-indica (L.) Mill., *Melaleuca citrina* Craven, *Galactites tomentosa* Moench) in 6 cases (15.4%), involved in predation activities within apiaries in 4 cases (10.3%), and collecting water in 2 cases (5.1%) (Figure 2).

### 3.2. Predation Pressure of the European Hornet on Hives

In apiary no. 2, composed of 15 hives, the total number of bees hunted by hornet workers was 21 during 8 h of observation. Thus, the average number of hornet workers per beehive was 1.4. Out of the 21 hornet attacks recorded in that location, 19 were against forager bees while returning to the hive and two against a bee at the hive entrance. In one of the latter two cases, the bee colony responded by balling (Appendix A). Differently, in apiaries no. 1 and 6, the hornet queen was never observed hunting honey bees during the 8 h observation period.

### 3.3. Observations on European Hornet Free-Living Colonies

#### 3.3.1. Hornet Ethogram

The different behaviours registered during our observations are presented in the following hornet ethogram implemented by behaviours reported in the literature for the genus *Vespa*.

*Ventilatory activity*. At the hornet nest entrance, some hornet workers were observed beating their wings in synchrony with each other at regular intervals, not touching each other, and with their heads facing outwards, thus creating a flow of air (Appendix A). This behaviour was likely only for nest thermoregulation. In fact, in *V. orientalis*, ventilating workers are not related to sentinel workers that inspect individuals entering the nest [25]. Furthermore, young queens are only seldom seen ventilating, whereas male hornets never do this type of activity [25]. In general, no ventilatory activity takes place inside the nest, e.g., upon the brood combs [25].

*Interaction with other species*. In the nest context, the interactions with other species taken into consideration regarded agonistic interactions, such as predation, in which individuals of *V. crabro* could be involved as predator or prey, or the interspecific competition for resources with individuals of the other species that entered the hornet nest (Appendix A). In the apiary context, the interactions with other species regarded the predation activity by hornets on honey bees. Usually, the hornet strategy is to attack the foraging bees returning to the hive in flight, whereas it rarely attacks bees at the hive entrance, probably because of the effective response of bees based on balling [15] (Appendix A). Baracchi et al. [15], investigating the attack strategy of *V. crabro* and the defence strategy of *A. mellifera*, highlighted that the European hornet is a mild predator, as it rarely attacks hives but rather prefers to prey upon honey bee foragers in flight, in accordance with our findings. The response of *A. mellifera* is usually exerted through bee-carpets and balling, which are considered to be effective also in view of the coevolution process of the species. 

*Food foraging activity and pellet preparation inside the nest.* The food collected by hornet workers or the queen during foraging activity can be liquid or solid. The former is mostly carbohydrate food, whereas the latter is protein food [7,8]. The main carbohydrate sources are tree sap, ripe fruits, honeydew, and flower nectar [7,8]. The solid protein food consists mainly of prey, with the European hornet usually forming a ball of meat in the hunting area after capture [15,26]. Less frequently, the whole prey or part of it is processed directly in the nest by hornets (Appendix A).

*Trophallaxis*. The exchange of food between two or more nestmates can take place between hornet adults or between adults and larvae. In particular, the larvae disgorge a secretion that adults use as a food source [8]. This larval saliva is highly attractive to adults and contains amino acids similar to those found in the nectar and it is essential for the survival of the colony [27,28]. Therefore, considering that wasps do not store food in the nest, this behaviour might have influenced the evolution of social life in these Hymenoptera [27,28]. In addition, the fifth-instar hornet larvae urge the adults to provide them with food by producing a sound resulting from the scraping of their mandibles against the inner wall of their cells [8,29,30,31] (Appendix A).

*Grooming*. Grooming behaviour of an insect is defined as the cleaning and removal of ectoparasites or other particles from its own body (self-grooming) or by a nest-mate (allo-grooming). Allo-grooming is common in eusocial insects and is only occasionally observed in solitary insect species [32]. In this study, allo-grooming between nest mates was observed 15 times, as shown in Table 2.

*Nest building and maintenance*. The materials that hornets collect to build their nests consist mostly of xylem wood or pieces of dead wood and, sometimes, the bark of live trees [11,12,14]. These materials are masticated and mixed with oral secretions. The nest of *V. crabro* is composed of plant fibre (70–75%) and saliva (20–25%) [33]. The salivary secretions are useful to keep the nest intact, protecting it from adverse conditions [33] (Appendix A).

*Undertaking behaviour*. Undertaking behaviour is an essential adaptation to social life, being important for colony hygiene in enclosed nests to prevent disease spread [34]. Social insects dispose of dead individuals in various ways (e.g., corpse removal, burial, and nestmate necrophagy) to avoid further contact between corpses and living members in a colony [34]. Removal of dead or dying larvae by workers was observed eight times in site no. 4 (Table 2).

*Cannibalism*. Under certain conditions like areas with a high wasp density and without enough prey or to supplement the diet with proteins, wasps can show cannibalism behaviour by feeding on conspecifics (adults or brood) [35]. Furthermore, cannibalism induced by artificial diet in captive conditions was observed in European hornet workers [28]. In our study, cannibalism events were directed only towards larvae (Table 2).

*Defecation*. The workers of *V. crabro* defecate under their nests and the faecal volatiles act as male attractants [36].

*Antennation over the capped brood cells*. The queen repeatedly touches the capped brood with its antennae.

#### 3.3.2. Intranidal and Extranidal Activity of the European Hornet Queen

In nest no. 1 (Cugnana), in which observations were carried out for five days, the queen left and returned to the nest 87 times. It spent ~70% of the total time of observation (40 h) outside the nest, engaged in predation activity or collecting various materials (Figure 2). In nest no. 6 (Telti), in which observations were carried out for one day, the queen left and returned to the nest 21 times, spending ~66% of the total time of observation (7 h) outside the nest, engaged in predation activity or collecting various materials (Figure 3).

For the queen at nest no. 1, the average duration (± SE) of each flight was 21.89 ± 1.21 min, whereas the mean time (±SE) spent inside the nest between two consecutive flights was 5.81 ± 0.62 min. The queen at nest no 6 spent on average (±SE) 19.67 ± 1.78 min for each trip, whereas the mean time (±SE) spent inside the nest between two consecutive flights was 6.85 ± 1.12 min. Furthermore, for this latter variable, the differences between the two queens were not statistically significant (time spent inside the nest between two consecutive trips: U = 752.5, N1 = 20, N2 = 86, *p* = 0.26). On the other hand, the duration of each trip differed significantly: U = 1547, N1 = 21, N2 = 87, *p* < 0.000.1.

Of all the time spent inside nest no. 1 during the first day of observation (~02:11, h:min), the queen dedicated 32.6% to nest construction, 26.6% to trophallaxis with larvae, 11.0% pellet food preparation, 0.12% to self-grooming, 0.09% to interaction with other species, and the remaining 29.5% on “other activities”, i.e., oviposition, walking, guarding, and resting. The observed frequencies of each behaviour observed during the first day of observation are reported in Table 2. The queen of site no. 6 spent a total of 143.80 min inside the nest, investing 35.6% of the time to trophallaxis with larvae, 21.1% to nest construction (building and maintenance), 8.1% to pellet food preparation, 4.7% to self-grooming, 0.8% to antennation on the capped cells, 0.3% to interaction with other species and, finally, the remaining 29.4%, on “other activities”, i.e., oviposition, walking, guarding, and resting. The number of events in a given time (frequency) observed for each behaviour are reported in Table 2.

In general, the daily activity of the queen began with the first flights at around 6:00 A.M. and lasted until 7:30/8:00 P.M., when observations were stopped. Interestingly, the queen’s foraging activity was not interrupted by the rain.

#### 3.3.3. Foraging Activity of the Colonies and Tasks Performed Inside the Nest by Workers

The daily activity of the free-living colonies of *V. crabro* located in S. Teodoro, Siniscola and Olbia S.M. (nests 3–5) were monitored in summer during the months of July, August, and September, respectively. In all three sites, the number of workers leaving and returning to the nest increased from 6:00 A.M. to 10:00 A.M., stabilized or had a slight decrease from 10:00 A.M. to 4:00 P.M., and decreased gradually between 4:00 P.M. and 8:00 P.M., with a slight increase from 6:00 P.M. and 8:00 P.M. in S. Teodoro (Figure 4). A total of 1780, 1346 and 14,390 worker foraging flights were recorded in the sites of S. Teodoro, Siniscola, and Olbia S.M., respectively. The behaviour of workers was homogeneous in all three sites, as workers returned to the nest without carrying anything with their jaws in most cases (on average, 75% of cases) (Figure 5). We also noticed that workers can bring unidentifiable food processed elsewhere (pellet) between their jaws (on average, 15% of cases), nest-building materials (on average, 8% of cases), or prey (on average, 1% of cases) (Figure 4). Overall, the proportion of foragers carrying anything to the nest was significantly the highest, whereas foragers carrying prey were the lowest (F_3,8_ = 34.93, *p* > 0.0001). We identified prey species in 39 cases (out of 124), with the most common prey being grasshoppers, followed by mantis and dragonflies (Figure 6). Identification of preyed species was not possible. The intranidal tasks performed by hornet workers in the Siniscola nest were mostly social interactions, such as trophallaxis between adults and nest building or maintenance. The observed frequencies of workers inside the nest are shown in Table 2.

## 4. Discussion

The main objective of our study was to verify whether *V. crabro* exhibited behavioural traits typical of an invasive process, with particular attention to predatory activity against *A. mellifera* in an area of new introduction. Although the European hornet is known to be a predator of various insects including bees [37], the results of our observations, conducted in three apiaries, confirm that *V. crabro* was only a mild predator of *A. mellifera* in north-eastern Sardinia as observed by Baracchi et al. [15] in its native area (Tuscany, central Italy).

Moreover, data recorded at the entrance of three hornet nests, revealed that only a very small fraction of prey species was brought intact or in pieces to the nest. In most cases, foraging hornets returned to the nest with meat pellets prepared in the hunting area. Contrary to what observed for *V. velutina* [38,39,40,41] and *V. crabro* under laboratory conditions [16], we observed no preference for honey bees compared to other insects in terms of individuals preyed upon and brought to the nest when prey was identifiable. This result was further confirmed by reports sent by citizens participating in the monitoring program from 2018 to 2020. In three years, the prevalent foraging activity of hornets was on fruits (20 cases), whereas predation on bees was reported only four times. Comparing our reports by citizen scientists (https://www.facebook.com/interregALIEMvespacrabroUNISS accessed on 15 December 2021) with similar studies on the Asian hornet *V. velutina* (https://www.stopvelutina.it/; https://www.vespavelutina.eu/it-it/ accessed on 15 December 2021), a clear relationship emerges between the latter species and its predation activity on honey bees in the context of the apiary. This finding is consistent with the negative impact that the Asian hornet has shown towards beekeeping in countries where it had been newly introduced [41]. In any case, further studies on the predation behaviour of *V. crabro* in Sardinian apiaries at different nest densities should be carried out.

The behaviour of *V. crabro* in Sardinia was closely similar to that described in its area of origin. In fact, in accordance with previous observations in its native area (eastern Asia in particular Japan and Taiwan) [8], our results showed that the queen invested significantly more time in extranidal than intranidal activities before the emergence of the workers. This behaviour normally changes when workers emerge, and the queen can devote herself exclusively to spawning and growing the colony [8]. The tasks performed by the queen within the nest at this stage of colony foundation were feeding/trophallaxis with larvae, nest construction/maintenance, and other activities like ovipositing, walking, and resting. In accordance with what was previously described in other hornet species [8,29,31,42], trophallaxis with the progeny was characterized by the emission of a sound by larvae to request food. The sound stopped either when the queen fed them or after the queen’s arrival at the nest (probably perceived by the hungry offspring through transmitted vibrations). Interestingly, antennation over the capped brood cells by the queen is a behavioural trait reported for *V. crabro* here for the first time. To our knowledge, there are no previous records of such queen-brood interaction in the literature. This observation suggests that the queen may somehow perceive stimuli from these cells and detect their stage of development. Finally, the lack of interruption of the extranidal activities of the queen due to rain was in accordance with previous observations of workers of European hornet [11]. 

The most interesting ethological aspect of hornet workers observed in this study is related to social behaviour among nest mates and the interactions with other species. Observations of trophallaxis showed that brief food exchange sessions were often followed by the exit of the individual that received the food from the nest, suggesting that there is a communication between mates about the source of food collected, as already observed in *Vespula germanica* [43,44]. Allo-grooming between nest mates was an additional social behaviour noted during observations. Based on our observations on the interactions between *V. crabro* and other species, one of the potential predators of *V. crabro* in Sardinia is Mauritanian tarantula *Tarentola mauritanica* (L.), also known as wall gecko, which succeeded in attacking one time out of three. It is interesting to note the interactions between the European hornet and other insects, such as the paper wasp *Polistes dominula* (Christ) (Hymenoptera: Vespidae) and the diurnal lepidoptera *Polygonia c-album* (L.) (Lepidoptera: Nymphalidae).

The observed activity of the *V. crabro* workers and queen, starting early in the morning at about 6:00 A.M., and ending at 8:00 P.M. at sunset, was in line with the observations of Matsuura [8]. Usually, *Vespula* and *Vespa* colonies have the highest level of activity in the morning [23]. The significant flow of workers coming out of the nest in the morning may be related to the urgent need of larvae for food after their nocturnal feast and/or to the fact that removing water in the form of dew is easier when collected early in the morning [11]. However, it is hard to compare our observations among different nests or with other studies because the time spent for diurnal foraging by hornets varies with species, colony development, observation period, and environmental conditions [23]. The influence of these parameters could explain the different daily flight patterns (6:00 A.M. to 8:00 P.M.) of hornet foragers from Siniscola, S. Teodoro, and Olbia S.M. nests. Nevertheless, the collection of fluids, such as nectar, tree sap, and water, is always the dominant foraging task performed by workers, as adults and larvae require carbohydrates [11]. Our observations on free-living nests confirmed this behaviour, as most foragers returned to the nest with apparently nothing between their mandibles, but probably carrying water or sugary exudates. Usually, hornets obtain liquid carbohydrates mostly by gnawing the bark of living trees to get sap [7,8]. During our study, citizen scientists provided evidence (photos or videos) of the collection of sugary liquids from trees, flowers, and waterholes, although most of the reports (51%) indicated the presence of *V. crabro* individuals on ripe fruit (primarily grapes and figs). It is interesting to note that, on the one hand, hornet workers can damage ripe fruit with their powerful mandibles [8], and, on the other hand, this social wasp can be a vector and natural reservoir of the yeast *Saccharomyces cerevisiae* Meyen ex E.C. Hansen, which is useful for the fermentation in winemaking, beer brewing, and bread process [45,46]. In addition, 17% of the observations of hornet workers on flowers reported by citizen scientists also suggested that this species could contribute to plant pollination, as observed by Thomson [47] for *Vespula pennsylvanica* (Saussure) (Hymenoptera: Vespidae). The role of wasps as pollinators is even known in Neotropical regions [48,49]. It was also observed that the wasps are attracted from flowers of small size, bulbous in shape with a wide entrance, and sugar-rich nectar [50]. The floral scent seems the main attraction factor [51]. In any case, until now, the ecological role of wasps is in general underestimated [52,53]. Finally, the observed ratio between predation activity (to feed the larvae) and sugary exudates collection may also have been influenced by the state of nest development, especially by the ratio between larvae and adults.

Currently, little information about *V. crabro* as an alien species is available and what there is derives from the USA. *Vespa crabro* is well established in North America after its release in the nineteenth century to protect forests from caterpillar outbreaks [2], as the European hornet is primarily a forest species. Its impact is limited to stripping bark from trees and ornamental plants [2,54]. In addition, no damage to beehives has been reported in the USA [54], in accordance with our observations.

## 5. Conclusions

*Vespa crabro* was accidentally introduced in Sardinia (Italy) in 2010. Despite the lack of native competitors, the favourable climate, and the availability of nesting sites (e.g., ancient olives groves) might have facilitated its spread in the northern area of the island, our observations showed no change in behavioural traits, compared to those described in its area of origin, which indicates that an invasive process has not occurred and is not likely to occur in Sardinia.

## Figures and Tables

**Figure 1 biology-11-00518-f001:**
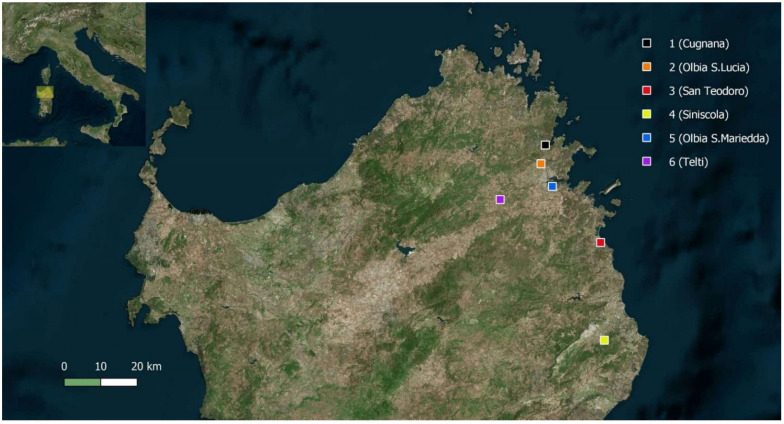
Map of the study area and location of nest sites.

**Figure 2 biology-11-00518-f002:**
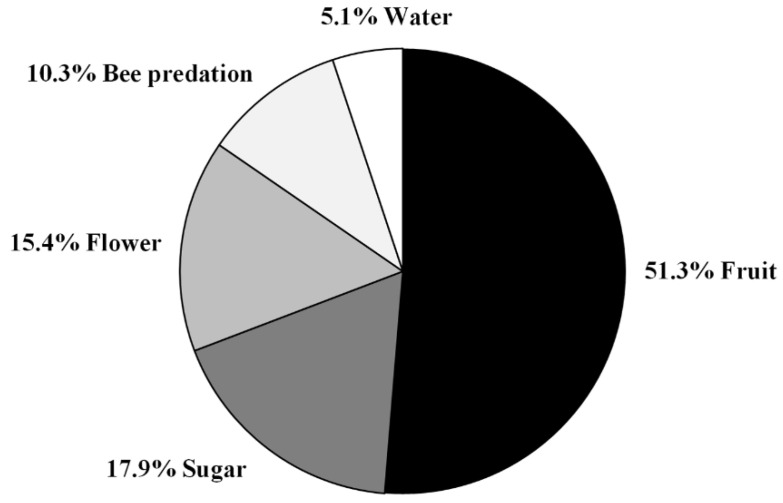
Relative frequency of European hornet foraging activity reports on ripe fruit, sugar exudates, flowers, water, and honey bees by citizen scientists.

**Figure 3 biology-11-00518-f003:**
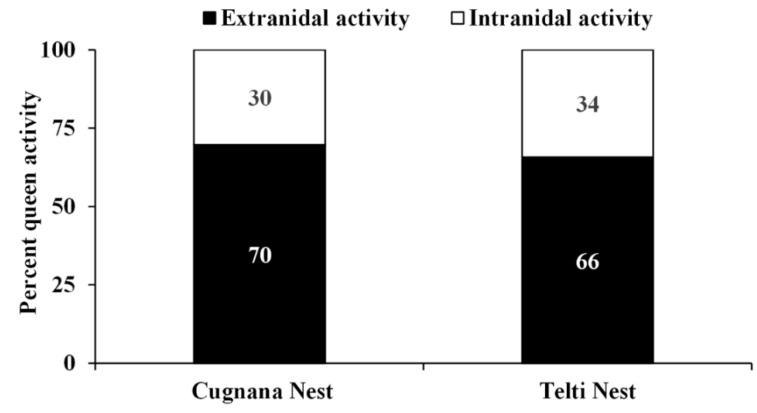
Relative frequency of extranidal and intranidal activity of European hornet queen on the total observation time (7 h) per nest (Cugnana and Telti nests, Sardinia, Italy). No significant differences were observed between the two nests (chi-squared = 0.47, df = 1, *p* = 0.49).

**Figure 4 biology-11-00518-f004:**
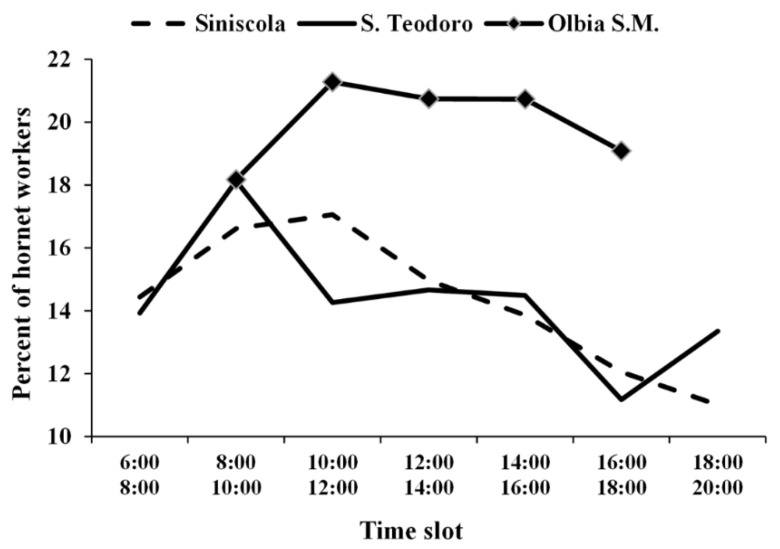
Weighted frequency per unit of time of European hornet flights during the day. Each point represents the average of the relative frequencies between hornets leaving the nest and those entering the nest.

**Figure 5 biology-11-00518-f005:**
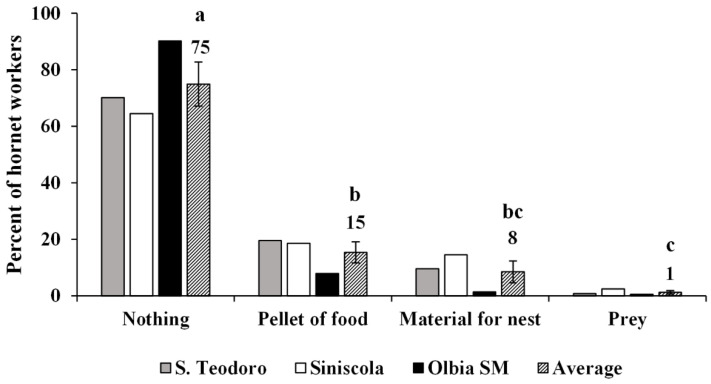
Relative frequency of European hornet workers entering the nest with nothing (possibly water or nectar), pellet of food, material for nest building, or prey between their mandibles in three nest sites. Different letters indicate significant differences among payloads (*p* < 0.0001).

**Figure 6 biology-11-00518-f006:**
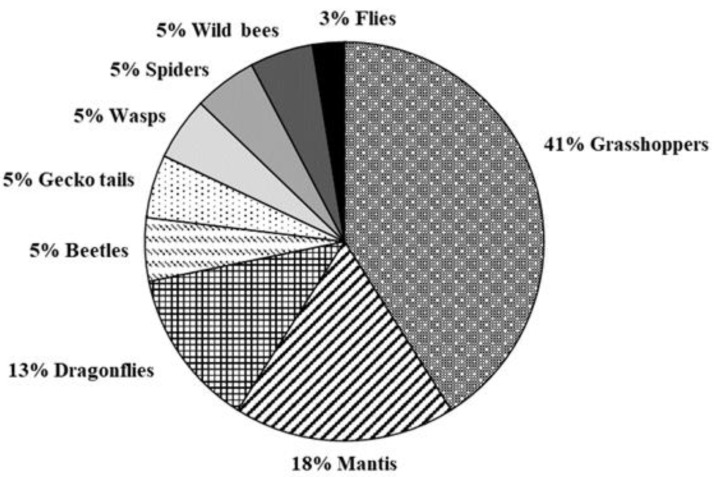
Relative frequency of the type of prey observed between the jaws of European hornet workers returning to the nest.

**Table 1 biology-11-00518-t001:** Features of the European hornet nests observed, and data collected in north-eastern Sardinia (Italy).

	Nest Number	1	2	3	4	5	6
Nest description	Social structure	Queen + brood	Queen + brood + workers	Queen + brood + workers	Queen + brood + workers	Queen + brood + workers	Queen + brood
	Habitat	Rural	Urban	Urban	Rural	Rural	Rural
	Context	Inside an apiary with 40 hives	Near an apiary with 15 hives (500 m)	Street (no hives)	Near an apiary with 20 hives (700 m)	Garden (no hives)	Inside an apiary with 24 hives
	Placement	Polystyrene hive	Tree crown	Inside a wall	Box-house	Hole in the tree	Beekeeping hive
	Locality	Cugnana (Sassari)	Olbia Santa Lucia (Sassari)	San Teodoro (Sassari)	Siniscola (Nuoro)	Olbia Santa Mariedda (Sassari)	Telti (Sassari)$
	Coordinates	41°00′37.1″ N 9°29′57.5″ E	40°57′51.40″ N 9°29′05.73″ E	40°46′08.82″ N 9°40′36.26″ E	40°31′40.66″ N 9°41′11.61″ E	40°54′29.74″ N 9°31′17.29″ E	40°52′35.14″ N 9°21′05.96″ E
	Altitude (m a.s.l.)	9	77	40	158	6	329
	Development	Number of capped brood cells = 3 Number of uncapped brood cells = 23	Number of incoming foragers/h = 61.7 Number of outgoing foragers/h = 60.7	Number of incoming foragers/h = 80.6 Number of outgoing foragers/h = 80.4	Number of incoming foragers/h = 47.8 Number of outgoing foragers/h = 47.0	Number of incoming foragers/h =544 Number of outgoing foragers/h =539	Number capped brood cells = 4 Number of uncapped brood cells = 18
Nest observations	Start/end dates	4 June 2018 8 June 2018	9 July 2019 9 July 2019	27 July 2019 29 July 2019	26 August 2019 28 August 2019	11 September 2019 13 September 2019	22 June 2020 22 June 2020
	Duration (days)	5	1	3	3	3	1
	Duration (hours:min)	40:00	08:00	22:05	28:10	26:28	07:00
Collected data	Task performed inside the nest	x	-	-	x	-	x
	Daily foraging activity rate	x	x	x	x	x	x
	Predation pressure on hives	x	x	-	-	-	x

**Table 2 biology-11-00518-t002:** Number of events of intranidal tasks performed by queens and workers of *Vespa crabro* in north-eastern Sardinia (Italy). Data are relative to the activity of the queen observed for 7 h in the Cugnana and Telti nests, and the activity of the workers for a total of 28:10 (hours:min) in the Siniscola nest.

Behaviour	Locality and Actor		
	Cugnana (No. 1) Queen	Telti (No. 6) Queen	Siniscola (No. 4) Workers
Nest building and maintenance	8	5	639
Pellet food preparation	4	16	3
Trophallaxis with larvae	2	72	15
Trophallaxis between adults	-	-	871
Antennation on capped brood	0	18	0
Self-grooming	2	62	3
Allo-grooming	-	-	15
Interaction with other species	4	2	3
Defecation	0	0	496
Ventilation	0	0	1
Larval removal	0	0	8
Cannibalism	0	0	9

## Data Availability

The data presented in this study are available on reasonable request from the corresponding author.

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
