# Peer review of "Bio-Ethology of Vespa crabro in Sardinia (Italy), an Area of New Introduction"

_biology, 2022, doi:10.3390/biology11040518_

Round 1

Reviewer 1 Report

Dear colleagues,

I reviewed the manuscript “Bio-ethology of Vespa crabro in an area of new introduction” (Pusceddu et al.). This manuscript presents an interesting work about Vespa crabro biology and its invasion in Sardinia. Authors analyzed some biological aspects to see possible impacts on local species, particularly honey bee colonies. The manuscript is well written and easy to read.

I have some comments authors could use to improve their manuscript.

Methods section

- Line 89: How many citizens participate to the work?

Results section

- Line 178: authors wrote “citizen scientists”. I would prefer another word as only “citizens” or “volunteers”. It’s strange to read “scientist” in the context.

- Lines 195-256: there is no results. I don’t understand this section 3.3.1. For me these information need to be moved to methods section.

- Line 215: I’m not convinced by the idea. In general, defense mechanisms of bees (A. mellifera) in Europe are not so efficient.

- Line 240: I think there is an error with the reference. Authors used the reference 32 to give information about nest structures. However, this article is only about the grooming behavior.

- Line 261: The nest was observed only one day. Why? Is this duration sufficient to give the information? For me, not really.

Figures

- Table 1: why the number of days for observations is so small? Only one day seems too small for me.

- Figure 2: Could authors add statistical results on the figure please?

- Table 2: I’m sorry, but I don’t understand what all these numbers in the table mean. It’s unclear for me. Could authors be clearer in the legend to explain these data please?

- Figures 3 & 4: Could authors precise the Y axis significations please? Only writing “%” is not clear enough.

Discussion section

I’m surprised to see no discussion about crabro invasion in USA. There is no publication with data about this invasion? What are the impacts of crabro on the US continent?

- Line 377: Please, author could write “Vespula”

- Line 379: Please write in italic the species name.

Author Response

Dear colleagues,

I reviewed the manuscript “Bio-ethology of Vespa crabro in an area of new introduction” (Pusceddu et al.). This manuscript presents an interesting work about Vespa crabro biology and its invasion in Sardinia. Authors analyzed some biological aspects to see possible impacts on local species, particularly honey bee colonies. The manuscript is well written and easy to read.

I have some comments authors could use to improve their manuscript.

Methods section

- Line 89: How many citizens participate to the work?

Response: The sentence “A total of 179 citizens participated to the work” was added in the new version of the manuscript.

Results section

- Line 178: authors wrote “citizen scientists”. I would prefer another word as only “citizens” or “volunteers”. It’s strange to read “scientist” in the context.

Response: "citizen scientists" has been replaced by “volunteers” in the new version of the manuscript.

- Lines 195-256: there is no results. I don’t understand this section 3.3.1. For me these information need to be moved to methods section.

Response: The ethogram describes the behaviour of V. crabro that we have directly observed in free-living colonies. Many of the observed behaviours are also supplemented with videos and photos reported in the supplementary material. We therefore believe that the most appropriate collocation of this paragraph is in the Results section.

- Line 215: I’m not convinced by the idea. In general, defense mechanisms of bees (A. mellifera) in Europe are not so efficient.

Response: We further clarified this point by reporting findings of Baracchi et al. [15] and our findings.

- Line 240: I think there is an error with the reference. Authors used the reference 32 to give information about nest structures. However, this article is only about the grooming behavior.

Response: The reference number was changed from 32 to 33.

- Line 261: The nest was observed only one day. Why? Is this duration sufficient to give the information? For me, not really.

Response: In some cases, we had limitations to access the sites and limited time for carrying out behavioural observations, as beekeepers removed quickly the V. crabro nests for the potential harm to bees. However, the overall database acquired seems sufficiently extensive to us (16 days of observations for a total of 131 hours and 43 minutes). In addition, the observed variability in behaviour among nests was low, as shown in Figure 4 (Figure 5 in the revised version).

Figures

- Table 1: why the number of days for observations is so small? Only one day seems too small for me.

Response: See response to the previous question.

- Figure 2: Could authors add statistical results on the figure please?

Response: We added the following sentence in the caption of Figure 2: “Not significant differences were observed between the two nests (chi-squared = 0.47, df = 1, P = 0.49)”.

- Table 2: I’m sorry, but I don’t understand what all these numbers in the table mean. It’s unclear for me. Could authors be clearer in the legend to explain these data please?

Response: Data reported in Table 2 represent the number of events observed in 420 minutes of observation for the nests of Cugnana e Telti and the number of events observed in 1690 minutes for the nest of Siniscola. To better clarify this aspect, we replaced “Observed frequency” with “Number of events” in the caption of Table 1.

- Figures 3 & 4: Could authors precise the Y axis significations please? Only writing “%” is not clear enough.

Response: In the new version of the manuscript, the meaning of Y axis in Figures 3 and 4 (Figures 4 and 5 in the revised version of the manuscript) was clarified by adding “Percent of hornet workers”.

Discussion section

I’m surprised to see no discussion about crabro invasion in USA. There is no publication with data about this invasion? What are the impacts of crabro on the US continent?

Response: We agree with the reviewer. The following sentence was added in the Discussion:

“Currently, few information about V. crabro as an alien species is available and refers to USA. In fact, V. crabro is well established in North America after its release in the nineteenth century to protect forests by caterpillar outbreaks [2], being the European hornet primarily a forest species. Its damage remains limited to stripping bark from trees and ornamental plants [2,54]. In addition, no damage to beehives has been reported in USA [54], in accordance with our observations”.

The paper by Beggs et al. (2011) has been accordingly added in the reference list as reference no. 54.

- Line 377: Please, author could write “Vespula”

Response:V. germanica” was replaced by “Vespula germanica”.

- Line 379: Please write in italic the species name.

Response: Done.

Reviewer 2 Report

This is an interesting article reporting a citizen science-based research that investigated the potential invasive traits of V. crabro in an area of new introduction, with special regard to predatory activity against honey bees. Observations on the hornet foraging behaviour in open field highlighted a prevalent foraging activity on fruits and no preference for bees compared to other recognizable arthropods. Furthermore, behavioural and predatory observations of V. crabro near to apiaries showed that foragers returned to nests carrying prey in 1% of cases. V. crabro did not show invasive traits nor notable behavioural changes in a non-native compared to its native area, as the hornet confirmed to be a mild predator of honey bees. The European hornet is not likely to represent a threat for beekeeping in Sardinia.

I would suggest to add a map locating the study sites in the region.

Author Response

This is an interesting article reporting a citizen science-based research that investigated the potential invasive traits of V. crabro in an area of new introduction, with special regard to predatory activity against honey bees. Observations on the hornet foraging behaviour in open field highlighted a prevalent foraging activity on fruits and no preference for bees compared to other recognizable arthropods. Furthermore, behavioural and predatory observations of V. crabro near to apiaries showed that foragers returned to nests carrying prey in 1% of cases. V. crabro did not show invasive traits nor notable behavioural changes in a non-native compared to its native area, as the hornet confirmed to be a mild predator of honey bees. The European hornet is not likely to represent a threat for beekeeping in Sardinia.I would suggest to add a map locating the study sites in the region.

Response: We thank the reviewer for the positive comments on our manuscript. A map locating the study area in the region and the nest sites was added in the new version of the manuscript. 

Reviewer 3 Report

Manuscript biology-1586768 titled “Bio-ethology of Vespa crabro in an area of new introduction” has valued field observations on daily foraging activity of free-living colonies in northern Sardinia, Italy, using the recording technique. Ethological aspects of hornet workers observed in this study concerned social behavior and interactions among nest mates. Authors concluded that Vespa crabro is not a major threat to honeybees and the risk of becoming invasive in Sardinia seems unlikely. The manuscript is well written and flows well from start to end. However, it lacks records of environmental conditions during the study periods that may influence ethological aspects of the hornet. I recommend consideration for publication in Biology MDPI with minor revision. I added my notes on the attached PDF for authors’ revision.  

Author Response

Manuscript biology-1586768 titled “Bio-ethology of Vespa crabro in an area of new introduction” has valued field observations on daily foraging activity of free-living colonies in northern Sardinia, Italy, using the recording technique. Ethological aspects of hornet workers observed in this study concerned social behavior and interactions among nest mates. Authors concluded that Vespa crabro is not a major threat to honeybees and the risk of becoming invasive in Sardinia seems unlikely. The manuscript is well written and flows well from start to end. However, it lacks records of environmental conditions during the study periods that may influence ethological aspects of the hornet. I recommend consideration for publication in Biology MDPI with minor revision. I added my notes on the attached PDF for authors’ revision.

Response: We thank the reviewer for the positive comments and suggestions on our manuscript. Responses to all the notes reported in the PDF were included in the new version of the manuscript.